# Dealing with the Unknown:
# Pessimistic Offline Reinforcement Learning

**Jinning Li, Chen Tang, Masayoshi Tomizuka, Wei Zhan**
Department of Mechanical Engineering
University of California, Berkeley
United States
{jinning_li,chen_tang,tomizuka,wzhan}@berkeley.edu

**Abstract:** Reinforcement Learning (RL) has been shown effective in domains where the agent can learn policies by actively interacting with its operating environment. However, if we change the RL scheme to offline setting where the agent can only update its policy via static datasets, one of the major issues in offline reinforcement learning emerges, i.e. distributional shift. We propose a Pessimistic Offline Reinforcement Learning (PessORL) algorithm to actively lead the agent back to the area where it is familiar by manipulating the value function. We focus on problems caused by out-of-distribution (OOD) states, and deliberately penalize high values at states that are absent in the training dataset, so that the learned pessimistic value function lower bounds the true value anywhere within the state space. We evaluate the PessORL algorithm on various benchmark tasks, where we show that our method gains better performance by explicitly handling OOD states, when compared to those methods merely considering OOD actions.

**Keywords:** Offline Reinforcement Learning, Out-of-Distribution States

## 1 Introduction

Reinforcement learning (RL), especially with high-capacity models such as deep nets, has shown its power in many domains, e.g., gaming, healthcare, and robotics. However, typical training schemes of RL algorithms rely on active interaction with the environments. It limits their applications in domains where active data collection is expensive or dangerous (e.g., autonomous driving). Recently, offline reinforcement learning (offline RL) has emerged as a promising candidate to overcome this barrier. Different from traditional RL methods, offline-RL learns the policy from a static offline dataset collected without iterative interaction with the environment. Recent works have shown its ability in solving various policy learning tasks [1, 2, 3]. However, offline RL methods suffer from several major problems. One of them is distributional shift. Unlike online RL algorithms, the state and action distributions are different during training and testing. As a result, RL agents may fail dramatically after deployed online. For example, in safety-critical applications such as autonomous driving, overconfident and catastrophic extrapolations may occur in out-of-distribution (OOD) scenes [4].

Many prior works [5, 6, 7, 8, 9, 10] try to mitigate this problem by handling OOD actions. They discourage the policies to visit OOD actions by designing conservative value functions, or estimating the uncertainty of Q-functions. Although constraining the policy can implicitly mitigate the problem of state distributional shift, few works have adopted measures to explicitly handle OOD states during the training stage. In this work, we propose the Pessimistic Offline Reinforcement Learning (PessORL) framework to explicitly limit the policy from visiting both unseen states and actions. We refer to the states or the actions that are not included in the training data as the unseen states or the unseen actions.

Our PessORL framework is inspired by the concept of pessimistic MDP in [11], where the reward is significantly small for unseen state-action pairs. We aim to limit the magnitude of the value function at unseen states, so that the agent can avoid or recover from unseen states. It is then crucial to precisely detect OOD states and shape the value function at those states. Since prior methods on OOD actions are derived from a similar concept, we can adapt their approaches to handle OOD states. There are

5th Conference on Robot Learning (CoRL 2021), London, UK.

mainly two approaches in the literature. One is to estimate the epistemic uncertainty of Q-function and subtract it from the original Q-function to get a conservative Q-function [6, 7, 8, 9, 10]. The other is to regularize the Q-function during the learning process [5]. The first method is highly sensitive to the trade-off between the uncertainty estimation and the original Q-function [12, 13] and the quality of uncertainty estimation [14].

Therefore, we follow the second approach, and add a conservative regularization term to the policy evaluation step of PessORL to shape the value function. We prove that PessORL learns a pessimistic value function that lower bounds the true value function, and forces the policy to avoid or recover from out-of-distribution states and actions. We evaluate the PessORL algorithm on various benchmark tasks. The performance of our method matches the state-of-the-art offline RL methods. In particular, we show that, by explicitly handling OOD states, we can further improve the policy performance compared to those methods merely considering OOD actions.

## 2  Related Works

A big challenge for offline reinforcement learning methods is to deal with the problems caused by unvisited states or actions in training data, which is also known as distributional shift. In model-free offline reinforcement learning, some works used importance sampling to fill the gap between the learned policy and the behavior policy in the training dataset [15, 16, 17, 18, 19]. There are also many works constrained the learned policy to be similar to the behavior policy by explicit constraints in the training dataset [6, 20, 21, 22, 23], so that the agent can avoid out-of-distribution actions during test time. The work in [13] proposed a latent space to constrain the policy to avoid deviating from the training data support. One further step to make the agent avoid actions that may cause itself deviate from the training data support is to get a conservative value function and thus a conservative policy. The works in [7, 8, 6, 9, 10] estimate the uncertainty of the learned Q-function, and then directly subtract it from the Q-function to get a conservative Q-function. Another way to get a conservative Q-function is to regularize the Q-function in the optimization problem during the learning process [5]. In model-based reinforcement learning (MBRL), there are also many algorithms that constrain the exploitation in the environment with effective uncertainty estimation methods [24, 25, 26, 27, 28, 11]. It is considered to be mature and reliable to detect OOD actions and states by methods from MBRL. Most of the aforementioned methods focus on OOD actions but not have explicit mechanism to deal with OOD states. In this paper, we focus on OOD states and propose a method to learn a pessimistic value function by adding regularization terms when updating Q-functions, and follow the works in the MBRL domain to establish the module to detect OOD states in our algorithm.

## 3  Background

### 3.1  Offline Reinforcement Learning

Given a Markov decision process (MDP), an RL agent aims to maximize the expectation of cumulative rewards. The MDP is represented by a tuple $\mathcal{M} = (\mathcal{S}, \mathcal{A}, \mathcal{P}, r, \gamma)$, where $\mathcal{S}$ is the state space, $\mathcal{A}$ is the action space, $\mathcal{P} : \mathcal{S} \times \mathcal{A} \times \mathcal{S} \to [0, 1]$ is the transition function, $r : \mathcal{S} \times \mathcal{A} \to \mathbb{R}$ is the reward function, and $\gamma$ is the discount factor. Typical RL algorithms optimize the policy using experience collected when interacting with the environment. Unlike those online learning paradigms, offline-RL algorithms rely solely on a static offline dataset, denoted by $\mathcal{D} = \left\{ \left( \mathbf{s}_t^i, \mathbf{a}_t^i, \mathbf{s}_{t+1}^i, r_t^i \right) \right\}$.

In this work, we focus on dynamic-programming-based RL algorithms under the offline setting, where we extract a policy from a learned value function for the underlying MDP in the training data. Standard Q-learning method estimates an approximate Q-function parametrized by $\theta$, i.e. $\hat{Q}_\theta(\mathbf{s}_t, \mathbf{a}_t)$. In each iteration, the Q-function is updated as follows:

$$\hat{Q}_\theta^{k+1} \leftarrow \arg\min_Q \frac{1}{2} \mathbb{E}_{\mathbf{s},\mathbf{a},\mathbf{s}' \sim \mathcal{D}} \left[ \left( Q(\mathbf{s},\mathbf{a}) - \hat{\mathcal{B}}^\pi \hat{Q}_\theta^k(\mathbf{s},\mathbf{a}) \right)^2 \right] \quad \text{(policy evaluation)}, \qquad (1)$$

where $\hat{\mathcal{B}}^\pi$ is the empirical Bellman update operator defined as:

$$\hat{\mathcal{B}}^\pi \hat{Q}_\theta^k(\mathbf{s},\mathbf{a}) = r(\mathbf{s},\mathbf{a}) + \gamma \mathbb{E}_{\mathbf{a}' \sim \hat{\pi}^k(\mathbf{a}'|\mathbf{s}')} \left[ \hat{Q}_\theta^k(\mathbf{s}',\mathbf{a}') \right]. \qquad (2)$$

For discrete action space, we define $\hat{\pi}^k$ as the optimal policy induced by the learned Q-function, i.e. $\hat{\pi}^k(\mathbf{a}|\mathbf{s}) = \delta \left[ \mathbf{a} = \arg\max_{\mathbf{a}} \hat{Q}_\theta^k(\mathbf{s},\mathbf{a}) \right]$. In this case, $\hat{\mathcal{B}}^\pi$ collides into the Bellman optimality

operator. When the action space is continuous, we follow actor-critic algorithms to approximate the optimal policy by executing a policy improvement step after policy evaluation in each iteration:

$$\hat{\pi}^{k+1} \leftarrow \arg\max_{\pi} \mathbb{E}_{\mathbf{s}\sim\mathcal{D},\mathbf{a}\sim\pi(\mathbf{a}|\mathbf{s})} \left[ \hat{Q}_\theta^{k+1}(\mathbf{s},\mathbf{a}) \right] \quad \text{(policy improvement)}. \tag{3}$$

In the rest of the paper, we denote $\mathcal{E}(Q, \hat{\mathcal{B}}^\pi \hat{Q}_\theta^k) = \frac{1}{2}\mathbb{E}_{\mathbf{s},\mathbf{a},\mathbf{s}'\sim\mathcal{D}} \left[ \left( Q(\mathbf{s},\mathbf{a}) - \hat{\mathcal{B}}^\pi \hat{Q}_\theta^k(\mathbf{s},\mathbf{a}) \right)^2 \right]$ as the Bellman update error for simplicity.

## 3.2 Uncertainty-Based Methods and Pessimistic Value Functions

By observing Eqn. 1, it is obvious that the Q-function $\hat{Q}_\theta$ is never evaluated or updated at states or actions that never appear in the dataset. The agent may behave unexpectedly or unpredictably at those unseen states or actions during test time. For dynamic-programming-based approaches, one way to address the issue of unseen actions is to estimate the epistemic uncertainty of Q-function and subtract it from the original Q-function [6, 7, 8, 9, 10]. The uncertainty is estimated based on an ensemble of learned Q-functions, and the final conservative Q-function becomes $Q_{\mathrm{c}}(\mathbf{s},\mathbf{a}) = \mathbb{E}_{Q\sim\mathcal{P}_\mathcal{D}(Q)}[Q(\mathbf{s},\mathbf{a}) - \alpha\mathrm{Unc}(\mathcal{P}_\mathcal{D}(Q))]$, where Unc is defined to be some uncertainty estimation metric, and $\mathcal{P}_\mathcal{D}(Q)$ is the distribution over possible Q-functions. Because the uncertainty metric is directly subtracted, uncertainty-based methods is highly sensitive to the quality of uncertainty estimation. Meanwhile, it is difficult to find an ideal $\alpha$ to balance the original Q-function and Unc.

Another way is to regularize the Q-function at the step of policy evaluation. A representative example is Conservative Q-Learning (CQL) [5]. Assuming that the dataset $\mathcal{D}$ is collected with a behavior policy $\pi_\beta(\mathbf{a}|\mathbf{s})$, and $\hat{\pi}^k(\mathbf{a}|\mathbf{s})$ is the learned policy at iteration $k$, the policy evaluation step in CQL becomes:

$$\hat{Q}^{k+1} \leftarrow \arg\min_{Q} \alpha \left( \mathbb{E}_{\mathbf{s}\sim\mathcal{D},\mathbf{a}\sim\hat{\pi}^k(\mathbf{a}|\mathbf{s})}[Q(\mathbf{s},\mathbf{a})] - \mathbb{E}_{\mathbf{s}\sim\mathcal{D},\mathbf{a}\sim\pi_\beta(\mathbf{a}|\mathbf{s})}[Q(\mathbf{s},\mathbf{a})] \right) + \mathcal{E}(Q, \hat{\mathcal{B}}^\pi \hat{Q}_\theta^k). \tag{4}$$

In the rest of the paper, we denote $\mathcal{C}(Q)$ as the cost term adopted from the CQL, i.e., $\mathcal{C}(Q) = \alpha \left( \mathbb{E}_{\mathbf{s}\sim\mathcal{D},\mathbf{a}\sim\hat{\pi}^k(\mathbf{a}|\mathbf{s})}[Q(\mathbf{s},\mathbf{a})] - \mathbb{E}_{\mathbf{s}\sim\mathcal{D},\mathbf{a}\sim\pi_\beta(\mathbf{a}|\mathbf{s})}[Q(\mathbf{s},\mathbf{a})] \right)$. It is worth noting that the aforementioned methods all focus on OOD actions, but they do not have an explicit mechanism to deal with OOD states, which motivates us to develop the PessORL framework in this work.

# 4 Pessimistic Offline Reinforcement Learning Framework

In this section, we introduce the PessORL framework to mitigate the issue of state distributional shift. In particular, we propose a novel conservative regularization term in the policy evaluation step. It can then be integrated into Q-learning or actor-critic algorithm, which will be described in Sec. 5.

## 4.1 How To Deal With OOD States

Assuming the dataset $\mathcal{D}$ is collected with a behavior policy $\pi_\beta(\mathbf{a}|\mathbf{s})$, and the states $\mathbf{s}$ are distributed according to a distribution $d^{\pi_\beta}(\mathbf{s})$ in the dataset, we propose to solve the problem caused by state distributional shift by augmenting the policy evaluation step in CQL [5] with a regularization term scaled by a trade-off factor $\varepsilon$:

$$\min_{Q} \varepsilon \left( \mathbb{E}_{\mathbf{s}\sim d^\phi(\mathbf{s}),\mathbf{a}\sim\hat{\pi}^k(\mathbf{a}|\mathbf{s})}[Q(\mathbf{s},\mathbf{a})] - \mathbb{E}_{\mathbf{s}\sim d^{\pi_\beta}(\mathbf{s}),\mathbf{a}\sim\hat{\pi}^k(\mathbf{a}|\mathbf{s})}[Q(\mathbf{s},\mathbf{a})] \right) + \mathcal{E}(Q, \hat{\mathcal{B}}^\pi \hat{Q}_\theta^k) + \mathcal{C}(Q), \tag{5}$$

where $d^\phi(\mathbf{s})$ is a particular state distribution of our choice.

The idea is to use the minimization term $\varepsilon \left( \mathbb{E}_{\mathbf{s}\sim d^\phi(\mathbf{s}),\mathbf{a}\sim\hat{\pi}^k(\mathbf{a}|\mathbf{s})}[Q(\mathbf{s},\mathbf{a})] \right)$ to penalize high values at unseen states in the dataset, and the maximization term $\varepsilon \left( \mathbb{E}_{\mathbf{s}\sim d^{\pi_\beta}(\mathbf{s}),\mathbf{a}\sim\hat{\pi}^k(\mathbf{a}|\mathbf{s})}[Q(\mathbf{s},\mathbf{a})] \right)$ to cancel the penalization at in-distribution states. The regularized Q-function could then push the agent towards regions close to the states from the dataset, where the values are higher. To achieve it, we need to find a distribution $d^\phi(\mathbf{s})$ that assigns high probabilities to states far away from the dataset, and low probabilities to states near the training dataset. We will instantiate a practical design of $d^\phi(\mathbf{s})$ in Sec. 5. For now, we just assume $d^\phi(\mathbf{s})$ assigns high probabilities to OOD states.

## 4.2 Theoretical Analysis

In this section, we analyze the theoretical properties of the proposed policy evaluation step. The proof and more details can be found in Appendix A.

We define $k \in \mathbb{N}$ as the iteration of policy evaluation, i.e. $\hat{Q}^k$ denotes the optimized Q-function in the $k-$th iteration obtained by PessORL. $Q^\pi$ is defined to be the true Q-function under a policy $\pi(\mathbf{a}|\mathbf{s})$ in the underlying MDP without any regularization. The true Q-function can be written in a recursive form via the exact Bellman operator, $\mathcal{B}^\pi$, as $Q^\pi = \mathcal{B}^\pi Q^\pi$. We define $\hat{V}^k$ as the value function under a policy $\pi(\mathbf{a}|\mathbf{s})$, $\hat{V}^k(\mathbf{s}) = \mathbb{E}_{\mathbf{a} \sim \pi(\mathbf{a}|\mathbf{s})}[\hat{Q}^k(\mathbf{s}, \mathbf{a})]$. For the true value function $V^\pi$ in the underlying MDP, we also have $V^\pi = \mathcal{B}^\pi V^\pi$.

We first introduce the theorem that the learned value function is a lower bound of the true one without considering the sampling error defined in the Lemma A.1.

**Theorem 4.1** *Assume we can obtain the exact reward function $r(\mathbf{s}, \mathbf{a})$ and the transition function $T(\mathbf{s}'|\mathbf{s}, \mathbf{a})$ of the underlying MDP. Let $\hat{V}^\pi(\mathbf{s}) = \lim_{k \to \infty} \hat{V}^k(\mathbf{s})$. Then $\forall \mathbf{s} \in \mathcal{S}$, the learned value function via Eqn. 5 is a lower bound of the true one, i.e., $\hat{V}^\pi(\mathbf{s}) \leq V^\pi(\mathbf{s})$, if the ratio $\dfrac{\varepsilon}{\alpha}$ satisfies*

$$\frac{\varepsilon}{\alpha} \leq \min_{\mathbf{s}} \left( \sum_{\mathbf{a}} \pi(\mathbf{a}|\mathbf{s}) \left[ \frac{\pi(\mathbf{a}|\mathbf{s})}{\pi_\beta(\mathbf{a}|\mathbf{s})} - 1 \right] \right) \left( \frac{|d^\phi(\mathbf{s}) - d^{\pi_\beta}(\mathbf{s})|}{d^{\pi_\beta}(\mathbf{s})} \sum_{\mathbf{a}} \frac{\pi^2(\mathbf{a}|\mathbf{s})}{\pi_\beta(\mathbf{a}|\mathbf{s})} \right)^{-1}.$$

It is worth noting that the learned value function still lower bounds the true value function for any state and action in the training datasets, i.e. $\mathbf{s}, \mathbf{a} \in \mathcal{D}$, even when we consider the sampling error defined in the Lemma A.1. Further details are shown in Corollary A.1. We have no reward or transition pair collected at unseen states or actions outside the training dataset, so it is impossible to bound the error outside the training dataset when consider the sampling error introduced by the reward function and the transition function.

We can now step further and show that the values at OOD states are lower than those at in-distribution states based on the learned value function. The proof can be found in Appendix A.3.

**Theorem 4.2** *For any state $\mathbf{s} \in \mathcal{S}$, if $\varepsilon > 0$ is sufficiently large, then the expectation of the learned value function via Eqn. 5 under the state marginal $d^{\pi_\beta}(\mathbf{s})$ in the training data is higher than that under $d^\phi(\mathbf{s})$, i.e., $\mathbb{E}_{\mathbf{s} \sim d^\phi(\mathbf{s})}[\hat{V}^\pi(\mathbf{s})] < \mathbb{E}_{\mathbf{s} \sim d^{\pi_\beta}(\mathbf{s})}[\hat{V}^\pi(\mathbf{s})]$.*

During training time, we can at least evaluate Q-values of OOD actions based on in-distribution states. However, there is actually no information about immediate rewards at OOD states, thus no information about Q-values. Intuitively, under offline settings, the best we can do to mitigate the problem of OOD states is to suppress values at these OOD states, and raising values at in-distribution states, so that the agent can be attracted to the area where it is familiar near the training data. Thm. 4.2 indeed tells us PessORL models a value function that assigns smaller values to OOD states compared to those at in-distribution states. Optimizing a policy under such a value function is similar to forcing the policy to avoid unknown states and actions.

In summary, PessORL can learn a pessimistic value function that lower bounds the true value function. Furthermore, this value function assigns smaller values to OOD states compared to those at in-distribution states, which helps the agent avoid or even recover from OOD states.

## 5 Implementing the Algorithm

In this section, we introduce a practical PessORL algorithm based on Eqn. 5. This algorithm simply modifies the policy evaluation step of Deep Q-Learning or Soft Actor-Critic algorithms, which is easy to implement.

### 5.1 Detecting OOD states

In prior to designing the algorithm, we need to choose a proper $d^\phi(\mathbf{s})$, which requires a tool for OOD state detection. Following [11, 14, 29], we use bootstrapping to detect OOD states. In

particular, we train a bag of Gaussian dynamics models [29] $\{\hat{P}_1, \hat{P}_2, \ldots, \hat{P}_n\}$ where each model is $\hat{P}_i(\cdot|\mathbf{s}, \mathbf{a}) = \mathcal{N}(\mathbf{s} + \hat{f}_{\phi_i}(\mathbf{s}, \mathbf{a}), \hat{\Sigma}_{\phi_i})$. The function $\hat{f}_{\phi_i}$ outputs the mean difference between the next state and the current state, and $\Sigma_{\phi_i}$ models the standard deviation. OOD states are detected by estimating the uncertainty of bootstrap models at a given state $\mathbf{s} \in \mathcal{S}$. Concretely, we define

$u_\pi(\mathbf{s}) = \mathbb{E}_{\mathbf{a} \sim \pi(\mathbf{a}|\mathbf{s})} \left[ \frac{1}{n} \sum_{i=1}^{n} \left( \hat{f}_{\phi_i}(\mathbf{s}, \mathbf{a}) - \bar{f}_\phi(\mathbf{s}, \mathbf{a}) \right)^2 \right]$, where $\bar{f}_\phi(\mathbf{s}, \mathbf{a}) = \frac{1}{n} \sum_{i=1}^{n} \hat{f}_{\phi_i}(\mathbf{s}, \mathbf{a})$ is the

mean of outputs of all $\hat{f}_{\phi_i}$, and the actions are drawn from a policy distribution $\pi$. A high $u_\pi(\mathbf{s})$ value indicates the state is more likely to be an unseen state. Given a set of sampled states $\{s_1, s_2, \ldots, s_n\}$,

we can define a discrete distribution over it using $u_\pi(\mathbf{s})$: $\zeta(\mathbf{s}_i) = \dfrac{u(\mathbf{s}_i)}{\sum_j u(\mathbf{s}_j)}, i = 1, 2, ..., n$, which

assign high probabilities to OOD states. In the following section, we will use it to construct the distribution $d^\phi(\mathbf{s})$.

## 5.2 Practical Implementation of PessORL

We now introduce a practical PessORL algorithm. In practice, to obtain a well-defined distribution $d^\phi$, we add an additional optimization problem over $d^\phi$ into the original optimization problem. The resulting optimization problem for the policy evaluation step is:

$$\min_Q \max_{d^\phi} \left[ \varepsilon \left( \mathbb{E}_{\mathbf{s} \sim d^\phi(\mathbf{s}), \mathbf{a} \sim \hat{\pi}^k(\mathbf{a}|\mathbf{s})} [Q(\mathbf{s}, \mathbf{a})] - \mathbb{E}_{\mathbf{s} \sim d^{\pi_\beta}(\mathbf{s}), \mathbf{a} \sim \hat{\pi}^k(\mathbf{a}|\mathbf{s})} [Q(\mathbf{s}, \mathbf{a})] \right) + \mathcal{R}(d^\phi) \right]$$
$$+ \mathcal{E}(Q, \hat{\mathcal{B}}^\pi \hat{Q}_\theta^k) + \mathcal{C}(Q), \quad (6)$$

where $\mathcal{R}(d^\phi)$ is a regularization term inspired by [5] in order to stabilize the training. If we choose $\mathcal{R}(d^\phi) = -D_{\mathrm{KL}}(d^\phi(\mathbf{s}) \,||\, \zeta(\mathbf{s}))$, where $\zeta(\mathbf{s})$ is the distribution we obtained from uncertainty estimations, then $d^\phi(\mathbf{s}) \propto \zeta(\mathbf{s}) \exp \left( V^{\hat{\pi}^k}(\mathbf{s}) \right)$, where $V^{\hat{\pi}^k}(\mathbf{s}) = \mathbb{E}_{\mathbf{a} \sim \hat{\pi}^k(\mathbf{a}|\mathbf{s})} [Q(\mathbf{s}, \mathbf{a})]$. The resulting $d^\phi$ is intuitively reasonable, because it assigns high probabilities to OOD states with high uncertainty estimations. In particular, $d^\phi$ assigns higher probabilities to states with high values, because we expect to penalize harder on them than those with low values already. With this choice of $d^\phi$ in

---

**Algorithm 1:** Pessimistic Offline Reinforcement Learning (PessORL)

---

1 **Initialize**: A Q network $Q_\theta$ parametrized by $\theta$, A target network $Q_{\bar{\theta}} = Q_\theta$ parametrized by $\bar{\theta}$, a policy network $\pi_\varphi$ parametrized by $\varphi$, and a bag of dynamics models $\{\hat{P}_1, \hat{P}_2, \ldots, \hat{P}_n\}$ to detect OOD states;

2 **// Dynamics Models Training** (Models are used by $u_{\pi_\varphi}(\mathbf{s})$ to detect OOD states in the policy evaluation step)

3 **for** *step i in range(0, M)* **do**

4     Train dynamics models $\{\hat{P}_1, \hat{P}_2, \ldots, \hat{P}_n\}$ according to the transitions in the dataset $\mathcal{D}$, so that we can later obtain an uncertainty estimation model $u_\pi(\mathbf{s})$ in the policy evaluation step;

5 **end**

6 **// Policy Evaluation and Improvement**

7 **for** *step t in range(0, N)* **do**

8     Update $Q_\theta$ according to Eqn. 7 with learning rate $\epsilon_\theta$ and $u_{\pi_\varphi}(\mathbf{s})$:

9     $\theta_t \leftarrow \theta_{t-1} + \epsilon_\theta \nabla_\theta J(\theta)$ ;

10     Update $\pi_\varphi$ according to the soft actor critic style objective and learning rate $\epsilon_\varphi$:

11     $\varphi_t \leftarrow \varphi_{t-1} + \epsilon_\varphi \mathbb{E}_{\mathbf{s} \sim d^{\pi_\beta}(\mathbf{s}), \mathbf{a} \sim \pi_\varphi(\mathbf{a})} [Q_\theta(\mathbf{s}, \mathbf{a}) - \log \pi_\varphi(\mathbf{a}|\mathbf{s})]$;

12     **if** *t mod target_update == 0* **then**

13        Soft Update the target network $\bar{\theta}_t \leftarrow (1 - \tau)\bar{\theta}_{t-1} + \tau\theta_{t-1}$

14     **end**

15 **end**

---

Eqn. 6, we obtain the following PessORL policy evaluation step:

$$\min_Q J(Q) = \min_Q \varepsilon \left( \log \sum_{\mathbf{s}} \zeta(\mathbf{s}) \exp\left( V^{\hat{\pi}^k}(\mathbf{s}) \right) - \mathbb{E}_{\mathbf{s} \sim d^{\pi_\beta}(\mathbf{s})}[V^{\hat{\pi}^k}(\mathbf{s})] \right)$$
$$+ \mathcal{E}(Q, \hat{\mathcal{B}}^\pi \hat{Q}_\theta^k) + \mathcal{C}(Q). \quad (7)$$

The first term in Eqn. 7 is very similar to weighted softmax values over the state space. It penalizes the softmax value over the state space, but also considers the distances between sample points and the training data. The two terms following the trade-off factor $\varepsilon$ is actually trying to decrease the discrepancy between the softmax value over OOD states and the average value over in-distribution states. Intuitively, it should enforce the learned value function to output higher values at in-distribution states, and lower values at out-of-distribution states. The logsumexp term in Eqn. 7 also mitigates the requirement for an accurate uncertainty estimation $\zeta(\mathbf{s})$ over the entire state space. Only those states with high values contribute to the regularization.

The complete algorithm is shown in Algorithm 1. We include the version for continuous action space which requires a policy network here, and note that if the action space is discrete, then we no longer need a policy network but just an implicit policy based on the learned Q-function. We implement PessORL on top of CQL [5], with its default hyperparameters. We also apply Lagrangian dual gradient descent to automatically adjust the trade-off factor $\varepsilon$. During the training process of offline reinforcement learning algorithms such as CQL and PessORL, we only have access to the dataset $\mathcal{D}$ instead of $d^{\pi_\beta}(\mathbf{s})$ and $\pi_\beta(\mathbf{a}|\mathbf{s})$. Therefore, we follow the convention in reinforcement learning community and approximated all expectations by Monte Carlo estimation in Eqn. 7.

## 6 Experiments

We compare our algorithm to prior offline algorithms: two state-of-the-art offline RL algorithms BEAR [6] and CQL [5]; two baselines adapted directly from online algorithms, actor-critic algorithm TD3 [30] and DDQN [31]; and behavior cloning (BC). The TD3 baseline is applied when the action space is continuous, whereas DDQN is trained when the action space is discrete. We evaluate each algorithm on a wide range of task domains, including tasks with both continuous and discrete state and action space. All baselines are run with the default code and hyperparameters from the original repositories. In particular, we are interested in the comparison between our algorithm with CQL, because we essentially add an additional state regularization term to the original CQL framework.

### 6.1 Performance on Various Environments

**Pointmass Mazes**. The task for the agent in this domain is to learn from expert demonstrations to navigate from a random start to a fixed goal. The expert dataset, which contains around 1000 trajectories all from the same start point to the same goal, is collected by online trained RL policy. During the test time of offline RL algorithms, we reset the start to a random point in the state space and the goal to the same fixed point as the dataset. In this way, the performance of the agent at unseen states are evaluated.

Before showing the performance, we first check if the OOD states detection is accurate, and hence, if we can successfully penalize high values at unseen states in training datasets. We evaluate the effectiveness of the OOD states detection method based on the accuracy of uncertainty estimation in the environment Pointmass. Figure 1(b) and (c) are visualizations of the training datasets and estimated uncertainty $u_\pi(\mathbf{s})$, both of which have the coordinate systems the same as that in the map (figure 1(a)). We use different colors in figure 1(b) and (c) to represent different values at each point in the map. The uncertainty estimations tend to be high (yellow areas in Fig. 1(c)) in area where the state density is low (blue areas in Fig 1(b)), and vice versa. This trend empirically shows that our uncertainty estimations are reasonable. We can trust them to detect OOD states when training offline RL algorithms.

We include the learning curves in figure 1(d), in which we evaluate each algorithm based on 3 random seeds, and report the average return. The shaded area represents the standard deviation of each evaluation. As we can see in the figures, PessORL outperforms other baselines in both hard and super hard environments. PessORL benefits from the augmented policy evaluation step in Eqn. 7. The learned value function produces high values at areas that have low uncertainty estimations, and low

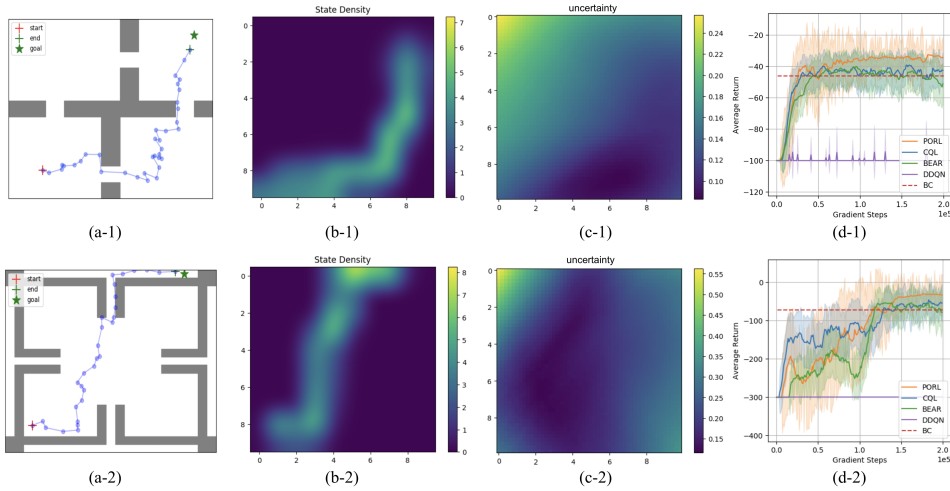

Figure 1: (a) The whole map of the environment; (b) The state density in training dataset; (c) The visualization of uncertainty estimation; (d) The learning curves. The top row (1) and the bottom row (2) are corresponding to PointmassHard-v0 and PointmassSuperHard-v0, respectively. We can see that almost all trajectories in the training datasets are located around the optimal trajectory from the start to the goal in the yellow areas in (b), indicating they are collected by a near-optimal policy.

values at highly uncertain areas (OOD states). Therefore, the agent can be "attracted" to the high value areas from low value and unfamiliar areas.

**Gym Tasks**. In this domain, we focus on the locomotion environments from MuJoCo, including Walker2d-v2, Hopper-v2, Halfcheetah-v2, and Ant-v2. Unlike Pointmass environment, we directly adopt the d4rl datasets [32] as our training data in the gym domains. We include four different types of datasets in our experiments, namely, "medium", "medium-expert", "random", and "expert". The "medium", "random", and "expert" dataset are all collected by a single policy, which is an either early-stopping trained, or randomly initialized, or fully trained expert policy. The "medium-expert" dataset is generated by mixing mediocre and expert quality data. We show the normalized scores averaged over 4 random seeds for all methods on gym domain in table 1. We directly ran all baselines

Table 1: Performance on Gym and Adroit Domains

| Domain | Task | BC | TD3 | BEAR | CQL | PessORL |
|--------|------|-----|------|------|-----|---------|
| **Gym** | hopper-medium | $29.71 \pm 2.43$ | $0.80 \pm 1.11$ | $50.08 \pm 4.49$ | $63.04 \pm 8.64$ | $\mathbf{76.39} \pm 4.80$ |
| | walker2d-medium | $13.06 \pm 1.74$ | $4.91 \pm 1.08$ | $34.20 \pm 8.57$ | $70.67 \pm 0.95$ | $\mathbf{75.95} \pm 2.96$ |
| | halfcheetah-medium | $35.81 \pm 0.96$ | $24.60 \pm 0.79$ | $41.25 \pm 2.45$ | $48.66 \pm 0.11$ | $\mathbf{49.21} \pm 0.59$ |
| | ant-medium | $83.42 \pm 9.63$ | $-63.55 \pm 0.28$ | $-45.78 \pm 3.18$ | $51.29 \pm 3.92$ | $\mathbf{85.31} \pm 8.56$ |
| | hopper-medium-expert | $85.22 \pm 1.38$ | $12.12 \pm 0.64$ | $37.67 \pm 1.84$ | $105.84 \pm 3.62$ | $\mathbf{112.80} \pm 4.30$ |
| | walker2d-medium-expert | $16.12 \pm 0.58$ | $4.77 \pm 2.06$ | $17.29 \pm 2.46$ | $75.27 \pm 13.33$ | $\mathbf{89.67} \pm 6.73$ |
| | halfcheetah-medium-expert | $\mathbf{37.04} \pm 2.77$ | $-1.41 \pm 0.99$ | $-2.93 \pm 4.54$ | $19.13 \pm 9.81$ | $24.33 \pm 12.24$ |
| | ant-medium-expert | $\mathbf{66.49} \pm 0.76$ | $-63.54 \pm 0.92$ | $31.31 \pm 8.72$ | $34.44 \pm 30.36$ | $49.95 \pm 16.76$ |
| | hopper-random | $9.44 \pm 1.41$ | $8.47 \pm 0.52$ | $9.29 \pm 2.58$ | $10.36 \pm 3.68$ | $\mathbf{10.82} \pm 4.11$ |
| | walker2d-random | $2.08 \pm 0.54$ | $\mathbf{6.04} \pm 1.08$ | $0.72 \pm 0.82$ | $1.11 \pm 5.01$ | $2.66 \pm 3.15$ |
| | halfcheetah-random | $2.25 \pm 0.45$ | $27.95 \pm 0.59$ | $2.16 \pm 0.28$ | $26.62 \pm 1.28$ | $\mathbf{28.58} \pm 0.96$ |
| | ant-random | $26.00 \pm 0.57$ | $-37.31 \pm 1.16$ | $24.05 \pm 2.42$ | $26.65 \pm 8.65$ | $\mathbf{27.86} \pm 4.87$ |
| | hopper-expert | $109.82 \pm 1.44$ | $1.81 \pm 1.04$ | $0.78 \pm 4.57$ | $104.41 \pm 10.26$ | $\mathbf{110.67} \pm 8.60$ |
| | walker2d-expert | $60.66 \pm 2.26$ | $-0.95 \pm 0.58$ | $21.37 \pm 3.52$ | $105.55 \pm 2.91$ | $\mathbf{109.37} \pm 4.94$ |
| | halfcheetah-expert | $\mathbf{94.22} \pm 1.09$ | $-1.40 \pm 2.85$ | $13.55 \pm 7.67$ | $80.19 \pm 14.55$ | $71.33 \pm 20.16$ |
| | ant-expert | $\mathbf{73.57} \pm 3.28$ | $55.07 \pm 1.99$ | $44.65 \pm 10.39$ | $59.04 \pm 20.66$ | $60.54 \pm 17.83$ |
| **Adroit** | pen-human | $34.46 \pm 0.83$ | $-3.83 \pm 0.43$ | $35.62 \pm 1.34$ | $54.49 \pm 7.58$ | $\mathbf{63.07} \pm 4.36$ |
| | door-human | $1.46 \pm 0.06$ | $-0.19 \pm 0.01$ | $-0.33 \pm 0.05$ | $1.86 \pm 0.29$ | $\mathbf{2.28} \pm 0.14$ |
| | hammer-human | $1.35 \pm 0.09$ | $0.26 \pm 0.12$ | $0.46 \pm 0.03$ | $3.91 \pm 0.34$ | $\mathbf{4.24} \pm 0.28$ |
| | relocate-human | $0.04 \pm 0.02$ | $-0.32 \pm 0.03$ | $-0.30 \pm 0.10$ | $0.15 \pm 0.04$ | $\mathbf{0.33} \pm 0.12$ |
| | pen-cloned | $23.58 \pm 1.14$ | $-3.91 \pm 0.36$ | $28.92 \pm 9.62$ | $35.23 \pm 11.03$ | $\mathbf{39.02} \pm 9.25$ |
| | door-cloned | $0.15 \pm 0.08$ | $-0.33 \pm 0.01$ | $-0.16 \pm 0.04$ | $\mathbf{1.72} \pm 0.14$ | $1.69 \pm 0.35$ |
| | hammer-cloned | $0.40 \pm 0.07$ | $0.25 \pm 0.04$ | $0.21 \pm 0.34$ | $0.53 \pm 0.52$ | $\mathbf{0.95} \pm 0.38$ |
| | relocate-cloned | $-0.24 \pm 0.11$ | $\mathbf{-0.14} \pm 0.08$ | $-0.23 \pm 0.13$ | $-0.28 \pm 0.57$ | $-0.26 \pm 0.24$ |

from their original repositories with their default parameters, and we only report the average scores we actually obtained. As we can see in the table, PessORL outperforms all other offline RL methods on a majority of tasks on gym domains. PessORL works especially well with mediocre quality datasets according to the results. In fact, it is one of the advantages of offline RL methods over behavior cloning on medium quality datasets, because offline RL methods take advantage of the information both from the reward and the underlying state and action distributions in training datasets, instead of simply imitating behavior policies as behavior cloning. Medium quality datasets are also considered to be similar to real-world datasets. Therefore, it is important for an offline RL method to perform well in medium quality datasets. We also note that PessORL shares some good properties with CQL, such as satisfying performance on mixed quality datasets. PessORL and CQL both outperform other offline methods on medium-expert datasets with PessORL better between them. The reason is that offline RL methods can "stitch" [32] different trajectories from different policies together according to the information from the reward.

**Adroit Tasks** The adroit domain [33] provides more challenging tasks than the Pointmass environment and the gym domain. The tasks include controlling a 24-DoF simulated Shadow Hand robot to twirl a pen, open a door, hammer a nail, and relocate a ball. Similar to the datasets in the gym domain, we also directly use the d4rl datasets as the training datasets in our experiments. The performance of PessORL and all baselines is shown in table 1. The normalized scores of all methods are average returns on 5 random seeds. We note that PessORL has better performance than other baselines on adroit domains. It is a great advantage for PessORL to learn useful skills from human demonstrations on these high dimensional and highly realistic robotic simulations.

## 6.2 Discussions and Limitations

The main contribution of this work is to explicitly limit the values at OOD states, so that the learned policy can act conservatively at OOD states and drives the agent back to the familiar areas near the training data. We are interested to see if our framework can indeed induce a different behavior on OOD states. We use $\Delta_k = \max_{\mathbf{s} \in \mathcal{S}}[V(\mathbf{s})] - \mathbb{E}_{\mathbf{s} \sim \mathcal{D}}[V(\mathbf{s})]$ as a metric to evaluate it at each iteration. If $\Delta_k$ is close to zero, then intuitively it indicates the values at OOD states are lower than those at in-distribution states. In Fig. 2, we plot $\Delta_k$ at each iteration in hopper-medium-v0. As is shown in

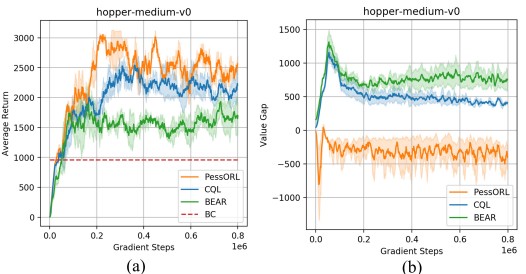

Figure 2: (a) The learning curves in hopper-medium-v0. (b) The discrepancy $\Delta_k$ as a function of gradient steps for PessORL, CQL, and BEAR.

the figure, PessORL successfully limits $\Delta_k$ to be non-positive, which meets our goal in this work and aligns with the statement in theorem 4.2.

On the gym domain, we notice that the performance of PessORL and CQL on datasets containing expert trajectories is not satisfying, often not as good as BC. We believe it is because of overly conservative value estimation. In fact, it is widely believed that conservative methods suffer from underestimation [14]. The conservative objective function in Eqn. 7 sometimes assign values that are too low to OOD states and actions. Besides, the uncertainty estimation method cannot be guaranteed to be precise on high-dimensional spaces. It is actually a possible future work direction to solve the underestimation and uncertainty estimation problems in conservative methods.

## 7 Conclusion

We propose a Pessimistic Offline Reinforcement Learning framework to deal with out-of-distribution states. In particular, we add a regularization term in policy evaluation step to shape value function, so that we can improve its extrapolation to OOD states. We also provide theoretical guarantees that the learned pessimistic value function lower bounds the true one and assigns smaller values to OOD states compared to those at in-distribution states. We evaluate the PessORL algorithm on various benchmark tasks, where we show that our method gains better performance by explicitly handling OOD states compared to those methods merely considering OOD actions.

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
