# OpenReview forum: "Dealing with the Unknown: Pessimistic Offline Reinforcement Learning"
_robot-learning.org/CoRL/2021/Conference — CoRL2021 Poster_

### Official Review · Reviewer_YDuP · 2021-07-23

**Originality:** Good
**Technical Quality:** Very Good
**Clarity Of Presentation:** Very Good
**Impact:** 3

**Recommendation:**

Weak Accept: I recommend accepting the paper, but will not argue for my recommendation if the majority of other reviewers have a different opinion.

**Summary:**

This paper proposes a method named PORL to handle OOD states and actions in the offline RL setting. But constructing an OOD state distribution, penalizing values from OOD distributions, the offline learned policy can avoid visiting OOD states, and as a result, show impressive performance in various domains.

**Issues:**

Already mentioned in "Strengths and weaknesses".

**Reviewer Expertise:**

Very good: Comprehensive knowledge of the area

**Strengths And Weaknesses:**

Strengths:
(1) It's a good to directly handle OOD states in offline RL policy evaluation and improvement, instead of implicitly handing OOD actions.
(2) It is reasonable to leverage uncertainty estimation to construct OOD state distributions. Experimental results also demonstrate its effectivenesses.
(3) Experimental results on the D4RL benchmark is promising.

Weaknesses:
(1) Since every states in the offline dataset are seen states, how can we say that some states are not seen when their estimated uncertainty is too high or exceeds a certain threshold? During the training procedure, every state sampled from the offline dataset is a seen state! To generate unseen states, you can use model-based methods such as MOPO.

(2) Theorem 4.1 requires that epsilon is small, while Theorem 4.2 requires that epsilon is sufficiently large. It seems like a conflict.

**Summary Of Recommendation:**

The motivation of this paper is reasonable and the results are promising. But the authors need to make some basic points more clear, e.g., what they refer to when they say "unseen states"?

---

> ### Author Response · Authors · 2021-08-25
> **Response to Reviewer YDuP**
>
> Thank you very much for your time to read our work thoroughly and your valuable advice!
> We are pleased that you found our work well-motivated and our results promising.
> Please find below our responses to your questions and suggestions.
>
> > *Since every states in the offline dataset are seen states, how can we say that some states are not seen when their estimated uncertainty is too high or exceeds a certain threshold? During the training procedure, every state sampled from the offline dataset is a seen state! To generate unseen states, you can use model-based methods such as MOPO.*
>
> We indeed generate states that are *not* included in the offline dataset during the training procedure, and the generating procedure is described in Appendix B from line 462 to 467.
> It is reasonable to use dynamics model learned by methods from model-based reinforcement learning domain, but to accelerate the training, we only adopt the method described in the paper.
>
> > *Theorem 4.1 requires that epsilon is small, while Theorem 4.2 requires that epsilon is sufficiently large. It seems like a conflict.*
>
> Thank you for pointing this out. We have added further details in Appendix A.4 to show that a feasible $\varepsilon$ exists.
>
> > *The authors need to make some basic points more clear, e.g., what they refer to when they say "unseen states"?*
>
> Thank you for your advice. We refer to the states that are not included in the training data as the unseen states. We have added the explanation in lines 33-35 in the revised paper.

---

> > ### Comment · Reviewer_YDuP · 2021-09-07
> > **Response to the authors' rebuttal**
> >
> > Thanks for the authors' detailed response, my questions have been resolved.

---

### Official Review · Reviewer_2d4v · 2021-07-23

**Originality:** Good
**Technical Quality:** Very Good
**Clarity Of Presentation:** Very Good
**Impact:** 4

**Recommendation:**

Strong Accept: I recommend accepting the paper and will argue for my recommendation even if other reviewers hold a different opinion.

**Summary:**

Offline reinforcement learning (RL) is an important topic, especially in the robotic community. In fact, in robotic applications, online  RL can be unsafe, as untrained policies can harm the system. Offline RL allows an expert (either a program or a human) to provide some demonstration of the task. The output of the RL algorithm would be, ideally, an agent that solves optimally the desired task leveraging on the suboptimal demonstrations.

The literature on the topic is rich. One of the main problems, is that the RL systems tends to make assumptions on unseen state-action pairs. When the policy moves towards actions that are not seen in the dataset (out of distribution OOD), the agent might end up in an unknown region of the state-action space. To avoid this, many algorithms attempt to provide a sort of regularization or constraint to avoid OOD actions.

The contribution of this paper is to introduce a similar regularization also on OOD states. The effect of such regularization term is to penalize OOD states, such that the policy avoids them.

To this end, the authors introduces a penalization term inspired by conservative $Q$-learning (CQL). Furthermore, they introduce a distribution that has high probability on OOD states, such that the penalization term is stronger in OOD states.

The paper provides both a theoretical analysis that shows that the value estimate is pessimistic for OOD states, and an empirical analysis that supports the claims of the paper.



**Issues:**

I'd be happy to understand why $d^\phi$ does not approximate the current state distribution (which seems to me more natural).

I would like to see Table 1 with confidence intervals.

**Reviewer Expertise:**

Very good: Comprehensive knowledge of the area

**Strengths And Weaknesses:**

__Strenghts__

The paper provides an effective algorithm to regularize OOD states. The idea seems to be original and well-motivated.
The regularization term seems to be effective (at least in the empirical analysis).

__Weaknesses__

In the first place, I am doubtful about the usage of $d^\pi$. In particular, I would not focus on assigning high probability to states that have high uncertainty, rather than to quantify the state-distribution induced by the policy (let's call it $d^\pi$).
In fact, this technique would penalize the states that are visited by the "current policy" but that are not contained in the dataset.
Quantify (or sampling from) $d^\pi$ can be done by sampling by the model $s_{t+1} = s_t + f(s_t, a_t)$ introcued in the paper, or using methods based on DICE for example [1].

Table 1 does not provide confidence intervals, therefore it is difficult to judge the statistical significance of the improvement.

[1] https://arxiv.org/pdf/1906.04733.pdf

**Summary Of Recommendation:**

The paper is, in my opinion, incremental. There is already a large work on offline reinforcement learning that focuses on OOD actions.
I still think that the paper is worth being accepted at the conference, as the method proposed is logical and original.


UPDATE
==

I have read the other reviews and the authors' responses. I am convinced that the authors did a good job in answering and incorporate the suggestions made.

---

> ### Author Response · Authors · 2021-08-25
> **Response to Reviewer 2d4v**
>
> Thank you very much for your detailed and insightful review!
> We appreciate it that you found our method effective and you liked our presentation style.
> Please find our responses to your questions and concerns below.
>
> > *I'd be happy to understand why $d^\phi$ does not approximate the current state distribution (which seems to me more natural).*
>
> In Sec. 5.1, we use bootstrapping to detect OOD states, and with this uncertainty estimation method, $d^\phi$ is then derived as described in Sec. 5.2.
> We focus on dealing with the problems caused by OOD states, but not on OOD state detection.
> Therefore, we intend to follow OOD state detection methods that have already been proved effective in prior reinforcement learning works [1, 2].
> It is reasonable to model the state distribution in the training data as $d^\phi$, by means of VAE [3] and so on. However, there is few prior works that adopt these methods to detect OOD states. Therefore, we are not sure about their performance.
>
>
> > *I would like to see Table 1 with confidence intervals.*
>
> Thank you very much for your advice. We have added confidence intervals to Table 1 in our revised paper.
>
> [1] R. Kidambi, A. Rajeswaran, P. Netrapalli, and T. Joachims. ''Morel:  Model-based offline reinforcement learning.'' 2020.
>
> [2] K. Chua, R. Calandra, R. McAllister, and S. Levine. ''Deep reinforcement learning in a handful of trials using probabilistic dynamics models.'' *Advances in Neural Information Processing Systems*, 2018.
>
> [3] Kingma, Diederik P., and Max Welling. ''Auto-encoding variational bayes.'' 2013.

---

> > ### Comment · Reviewer_2d4v · 2021-08-30
> > **Thank you**
> >
> > Thanks for your answers, and for updating the intervals in Table 1.
> > I will update my review accordingly.

---

### Official Review · Reviewer_DVzS · 2021-07-24

**Originality:** Very Good
**Technical Quality:** Very Good
**Clarity Of Presentation:** Very Good
**Impact:** 3

**Recommendation:**

Strong Accept: I recommend accepting the paper and will argue for my recommendation even if other reviewers hold a different opinion.

**Summary:**

One of the problems with offline RL is distribution shift, namely the fact that at evaluation time the agent visits parts of the state space which were not part of the training set. The authors propose a solution that builds on top of Conservative Q-Learning (CQL) and constructs a value function with high probability values inside the training dataset, and low probability for Out-Of-Distribution states. They achieve this by detecting OOD states based on their uncertainty and use that to construct a probability distribution $d_\phi$ with the abovementioned properties, that acts as a regularizer for the Q-function. The distribution $d_\phi$ heavily penalizes states based on two factors: their uncertainty estimate and their value estimate. This means that states with a higher value will be heavily penalized if their uncertainty is greater than zero.

**Issues:**

- How is the score normalized in table 1?
- There is a work that considers pessimistic initialization of the Q-function called OPIQ [1] and presented last year at ICLR. It doesn't explicitly consider OOD scenarios, but it is related to having a pessimistic value of the Q-function for unseen state-action pairs.
It could be of interest to show how PORL compares to that.
- It could also be interesting to see how PORL compares to other methods that use different types of OOD detection. Did you try some of the approaches mentioned in the related work?

[1] "Optimistic Exploration even with a Pessimistic Initialisation" Tabish Rashid, Bei Peng, Wendelin Boehmer, Shimon Whiteson, ICLR2020

**Reviewer Expertise:**

Fair: Some knowledge of the area

**Strengths And Weaknesses:**

The paper is well written and very clear. The experiments are well designed and many environments are presented.
The claims are supported by theoretical guarantees and by strong experimental performances against different baselines.
I would suggest including some references related to OOD detection based on states (see below).

**Summary Of Recommendation:**

This paper provides an intuitive solution to the problem of distribution shift in batch RL. The claims are well supported by theoretical and empirical evidence. For this reason, I recommend acceptance.

---

> ### Author Response · Authors · 2021-08-25
> **Response to Reviewer DVzS**
>
> Thank you very much for your time and valuable suggestions! We are very happy that you liked our presentation style and the design of our experiments. Our responses to your questions are listed as below.
>
> > *How is the score normalized in table 1?*
>
> We referred to the D4RL dataset [1] to normalize the score for each environment.
> The normalized score ranges from $0$ to $100$. A normalized score of $0$ corresponds to the averge return of an agent that takes uniformly random actions over the action space, whileas a score of $100$ corresponds to the average return of an expert in each domain.
> Therefore, the normalized score is calculated as:
>
> $\texttt{normalized score} = 100 * \dfrac{\texttt{score} - \texttt{random score}}{\texttt{expert score} - \texttt{random score}}$.
>
>
> > *There is a work that considers pessimistic initialization of the Q-function called OPIQ [2] and presented last year at ICLR. It doesn't explicitly consider OOD scenarios, but it is related to having a pessimistic value of the Q-function for unseen state-action pairs. It could be of interest to show how PORL compares to that.*
>
> This is really a good question.
> **[Updated] We appreciate your advice, and we have added experiments to compare OPIQ and PORL in Appendix C.2.
> Actually, the method to get a pessimistic Q-function in OPIQ is similar to that of the variant PessORL-unc in Appendix C.
> Both of them obtain pessimistic Q-functions by directly subtracting an uncertainty term from the learned Q-function.**
>
> > *It could also be interesting to see how PORL compares to other methods that use different types of OOD detection. Did you try some of the approaches mentioned in the related work?*
>
> The main focus of this paper is to solve the problems caused by OOD states, but not OOD state detection. Therefore, we do not intend to compare different OOD state detection methods in the paper.
> Nevertheless, we initially tried to use Random Network Distillation (RND) [3] to detect OOD states, but unfortunately that did not work well.
> Some of the uncertainty estimation methods in the related work come from a subfield of reinforcement learning, i.e., exploration.
> However, offline reinforcement learning generally requires a more accurate uncertainty estimation method than exploration, whereas it is considered to be mature and reliable to detect OOD states and actions by methods from model-based reinforcement learning [4].
> Therefore, we chose the OOD detection method described in Sec. 5.1.
>
> [1] J. Fu, A. Kumar, O. Nachum, G. Tucker, and S. Levine. ''D4rl: Datasets for deep data-driven reinforcement learning.'' 2020.
>
> [2] Tabish Rashid, Bei Peng, Wendelin Boehmer, Shimon Whiteson. ''Optimistic Exploration even with a Pessimistic Initialisation.'' ICLR2020.
>
> [3] Burda, Yuri, et al. ''Exploration by random network distillation.'' 2018.
>
> [4] S Levine, A Kumar, G Tucker, J Fu. ''Offline reinforcement learning: Tutorial, review, and perspectives on open problems.'' 2020.

---

> > ### Comment · Reviewer_DVzS · 2021-09-05
> > **Response to authors**
> >
> > I thank the authors for addressing all my questions, and I am glad they included the comparison to OPIQ. It was interesting to read their analysis and to have this additional experiment, I am sure it will be useful for future readers.

---

### Official Review · Reviewer_Qikv · 2021-07-30

**Originality:** Good
**Technical Quality:** Fair
**Clarity Of Presentation:** Good
**Impact:** 1

**Recommendation:**

Weak Reject: I recommend rejecting the paper, but will not argue for my recommendation if the majority of other reviewers have a different opinion.

**Summary:**

This submission proposed Pessimistic Offline Reinforcement Learning (PORL), an
offline-RL algorithm which addresses issues associated with out-of-distribution
(OOD) states, while previous work only focused on OOD actions.  PORL extends
prior work by incorporating a regularization term to lower the values
associated with unseen states, and which is inspired by a prior regularization
terms which was designed to lower the values associated with unseen actions.

**Issues:**

Major:

* Equations (1) and (2) do not describe Q-learning, as only the Bellman
  *evaluation* operator is used, rather than the Bellman *optimality* operator.
  Equations (1) and (2) will approximately lead to the value function of the
  agent which was used to perform the data collection.  However, there is no
  underlying assumption made in offline-RL (or in this paper) that the
  data-collecting agent is optimal or near-optimal.

* Why does $d^\phi(s)$ need to specifically be a distribution which assigns low
  probabilities to OOD states?  Can it not simply assign a uniform probability
  to all states?  In that case, the states present in the dataset will still
  have $d^{\pi_\beta}(s) > d^\phi(s)$, and the states not present in the
  dataset will still have $d^{\pi_\beta}(s) < d^\phi(s)$.  Isn't it possible
  that this would work well enough, without the need to learn the distribution
  $d^\phi(s)$?

* It is not obvious whether $\hat V^\pi(s) \leq V^\pi(s)$ implies $\hat
  Q^\pi(s, a) \leq V^\pi(s, a)$;  some discussion / proof should be provided.

* It is not clear how Theorem 4.2 implies that the PORL models a pessimistic
  MDP as formally defined in [10], as claimed in lines 163-164.  In fact, the
  formal definition of pessimistic-MDPs requires components, such as an
  additional absorbing state "HALT", which does not appear at all in this
  submission.  Finally, it is not even clear why this should matter.  The
  important property seems to simply be that $\hat V^\pi(s) < V^\pi$.  Why is
  it important that PORL learns values which are hypothetically associated with
  pessimistic MDPs?

* According to Theorem 4.1, $\varepsilon$ should be small enough to satisfy the
  inequality.  According to Theorem 4.2, $\varepsilon$ should be large enough
  to satisfy the equality in eq.(21) from the appendix.  However, no practical
  method is given to find a $\varepsilon$ which satisfies both conditions, or
  to even verify that a given $\varepsilon$ satisfies such conditions.  It is
  also not proven whether there can be a $\varepsilon$ which is both small
  enough and large enough to satisfy both conditions.

* "All baselines are run with the default code and hyperparameters from the
  original repositories."  Were these hyperparameters optimized for the same
  environments used in your evaluation?  It doesn't seem likely.  If so, this
  is a huge flaw in the evaluation.  How are the hyperparameters of PORL
  chosen?

Mid:

* The problems associated with distributional shift, OOD states and actions,
  could be described better, e.g., in lines 24-26, it is claimed that
  distributional shift might cause problems, but what kind of problems is not
  well described (yet).

* Neither $\mu$ nor its role are properly described.  It is particularly
  confusing that $\mu$ is described to match the state-marginal, while in Eq.4,
  it appears as the state-conditional "policy" $\mu(a\mid s)$.

* According to Eq.4, both CQL and PORL assumes access to the behavior policy
  $\pi_\beta$, which is a significantly stronger requirement compared to simple
  access to the dataset $\mathcal{D}$.  Or is it simply the case that the
  dataset actions (which are sampled according to the behavior policy) are used
  directly?  The details of how objective $J$ of Eq.7 is estimated should be
  provided (which expectations are computed exactly?  which are approximated
  using a Monte Carlo estimation).

* in lines 124 and 125, the statement "it is empirically impossible to derive a
  solution to maximize the KL divergence" needs a citation or further
  justification.

* In Eq.5, why only penalize a single action-value associated with unseen
  states, as opposed to all action-values associated with unseen states?  I.e.,
  why only use the action $a\sim\hat \pi^k(a\mid s)$?

* Theorem 4.1 contains confusing statements which switch between $Q$ and $V$.
  "For any action and any state ..., i.e. $\hat V^\pi(s) \leq V^\pi(s)$ ...".

* In Theorem 4.1, the definition of $\hat V^\pi$ was unclear.  I assume the
  intention was to define $\hat V^\pi$ as the convergence value $\hat V^\pi(s)
  = \lim_{k\to\infty} \hat V^k(s)$, but the way the sentence is worded it seems
  like there is an assumption that $\hat V^k$ converges to a specific value.

* A number of issues are present in the appendix, usually related to the
  authors skipping or not describing steps sufficiently.  Since this is the
  appendix, the authors should feel free to use enough space to be very clear,
  and complement the math with written descriptions of what is happening,
  without the burden of limited pages:

  * In line 384 of the appendix, "Setting the derivative to zero".  The
    derivative with respect to what?  It is not clear.  Also, how is the
    derivative of the two regularization terms computed?

  * In line 386 of the appendix, "It is obvious that the last two terms of Eqn.
    8, ..., is indefinite".  It is not obvious at all, and it is also unclear
    what the authors mean by "indefinite".  It cannot be a real number?  Since
    this is the appendix, the authors should feel free to use as much space as
    possible to be as clear as possible.

  * In line 400 of the appendix, "Hence, we have ... $\hat V^{k+1} \leq
    \mathfrak{B}^\pi \hat V^k$, which implies that each value-function update
    is a contraction."  I am not sure if this is a typo, but this does not seem
    to align with the formal definition of contraction, which is an operator
    which guarantees the outputs associated with any two inputs are closer to
    each other than the inputs to each other.  What do you mean by
    "contraction" here?

  * Eq 13 in the appendix suddenly mixes point-wise and vector notation.

Minor:

* PORL is an unfortunate acronym, since it conflicts with the widely used
  Partially Observable Reinforcement Learning (PORL).

* Calling Q-learning "dynamic-programming" (happens a few times) is inaccurate.

* Notation of policy in line 84 does not work for discrete actions;  should be
  $\delta\left[ a = \arg\max_a \ldots \right]$.

* $\mathcal{E}$ and $\mathcal{C}$ should be defined closer to their respective
  equations, i.e., i.e., $\mathcal{E}$ should be defined near Eq.1, and
  $\mathcal{C}$ should be defined near Eq.4.

* The notation is at times a little bit redundant, e.g., $s\sim\mathcal{D}$
  represents a state sampled from the dataset, and $s\sim d^{\pi_\beta}$
  represents the same exact thing.

* Not accurate to use the same operator $\mathfrak{B}^\pi$ for both
  state-values $V$ and action-values $Q$.

* Pessimistic MDPs should be formally described in the Background section.

**Reviewer Expertise:**

Good: General knowledge of the area

**Strengths And Weaknesses:**

Strengths:
* Extends similar methods for offline-RL in interesting and non-trivial ways.
* I strongly appreciate the focus on theory, rather than presenting only
  empirical results.

Weaknesses:
* Seems like the authors did not perform hyper-parameter selection during the
  evaluation, which is a huge red flag.
* A few key details are missing both in the theory and in the practical
  implementation (see Issues section).
* Some notation is confusing and/or not properly defined, or contains mistakes.
* Not very related to robot learning, and no robot experiments (not even
  simulated).

**Summary Of Recommendation:**

While I generally like the work and the main contribution of the submission, as
it stands, the submission has a number of flaws and issues which need to be
addressed.  They are not all equally important, and the two main ones which
really concern me are the lack of good hyperparameter selection (or a
description of how the PORL hyperparameters are chosen), and the questionable
relevance to the community of robot learning.

My current recommendation is based on the submission as it stands, and will
gladly update it based on the author response.

---

> ### Author Response · Authors · 2021-08-25
> **Response to Reviewer Qikv (Part 1/4)**
>
> (Please note that we split our response into four parts due to space constraints. Part 1/4)
>
> Thank you very much for your time to read our paper thoroughly and your valuable detailed review!
> We are pleased that you found our work interesting and that you appreciate the theoretical analysis. Please find below our responses to your concerns and suggestions.
>
> > *Not very relevant to the robot learning community.*
>
> In particular, one of our experimental environments is the Adroit environment, which consists of several manipulation tasks. It has been used as a platform in prior works of the robotic community [3] to study the application of RL on complex dexterous manipulation tasks. Also, CQL has been shown effective in some real robot experiments [4]. Since PessORL achieves better simulation performance than CQL in our experiments, we believe that PessORL has similar potential to be applied on real robots. More details can be found in Appendix D.
>
> **Major:**
>
> > *Equations (1) and (2) do not describe Q-learning, as only the Bellman evaluation operator is used, rather than the Bellman optimality operator. Equations (1) and (2) will approximately lead to the value function of the agent which was used to perform the data collection. However, there is no underlying assumption made in offline-RL (or in this paper) that the data-collecting agent is optimal or near-optimal.*
>
> Thanks for your advice. We modified the description in the Background section about the standard Q-learning and actor-critic algorithm to be more accurate. In particular, we make a note that the empirical Bellman operator collides into the Bellman optimality operator if we have discrete action space, as in standard Q-learning. If the action space is continuous, then we can only learn a policy to approximate the optimal policy. For clarification, the learned policy does not imitate the data-collecting agent. We optimize the policy by the policy improvement step in Eqn. 3. It could exceed the behavior policy in terms of return (the sum of rewards in each episode).
>
> > *Why does $d^\phi(s)$ need to specifically be a distribution which assigns low probabilities to OOD states? Can it not simply assign a uniform probability to all states? In that case, the states present in the dataset will still have $d^{\pi_\beta}(s) > d^{\phi}(s)$, and the states not present in the dataset will still have $d^{\pi_\beta}(s) < d^{\phi}(s)$. Isn't it possible that this would work well enough, without the need to learn the distribution $d^{\phi}(s)$?*
>
> Thank you very much for your advice.
> It is really a good point to assign $d^\phi (s)$ as a uniform distribution. It is worth to show the comparison between the $d^\phi (s)$ in the paper and the uniform distribution.
> **[Updated] We have added the comparison results to Appendix C.1.**
>
> > *It is not obvious whether $\hat{V}^\pi(s) \leq V^\pi(s)$ implies $\hat{Q}^\pi (s, a) \leq V^\pi(s, a)$; some discussion / proof should be provided.*
>
> For clarification, we did not intend to imply that we can lower bound the Q-function - we suppose you mean $\hat{Q}^\pi (s, a) \leq Q^\pi(s, a)$ by $\hat{Q}^\pi (s, a) \leq V^\pi(s, a)$ - from $\hat{V}^\pi(s) \leq V^\pi(s)$. In Appendix A.1, we discussed that the last two terms in Eqn. 8 could be either positive or negative, therefore, we could not simply lower bound the true Q-function $Q^\pi(s, a)$ point-wise by the estimated one $\hat{Q}^\pi$.
> We instead proved that the value function, which is the expectation of the Q-function, can be lower bounded.
> Then we obtained the result $\hat{V}^\pi(s) \leq V^\pi(s)$.

---

> > ### Author Response · Authors · 2021-08-25
> > **Response to Reviewer Qikv (Part 2/4)**
> >
> > (Continued. Part 2/4)
> >
> > > *It is not clear how Theorem 4.2 implies that the PORL models a pessimistic MDP as formally defined in [10], as claimed in lines 163-164. In fact, the formal definition of pessimistic-MDPs requires components, such as an additional absorbing state "HALT", which does not appear at all in this submission. Finally, it is not even clear why this should matter. It is not even clear why this should matter. The important property seems to simply be that $\hat{V}^\pi(s) < V^\pi(s)$. Why is it important that PORL learns values which are hypothetically associated with pessimistic MDPs?*
> >
> > In Sec. 4.2, we originally drew a connection between the property of our method PORL and that of a pessimistic MDP defined in the prior work [1]. We did not mean to formally formulate PORL into pessimistic-MDPs, but merely connect them in a high-level and intuitive way. To avoid confusion, we deleted the statement related to pessimistic-MDPs in Sec. 4.2. The key insight we would like to deliver in this section is that PORL can force the agent to avoid unknown or unfamiliar states and actions by shaping the value function.
> >
> > > *According to Theorem 4.1, $\varepsilon$ should be small enough to satisfy the inequality. According to Theorem 4.2, $\varepsilon$ should be large enough to satisfy the equality in eq.(21) from the appendix. However, no practical method is given to find a $\varepsilon$ which satisfies both conditions, or to even verify that a given $\varepsilon$ satisfies such conditions. It is also not proven whether there can be a $\varepsilon$ which is both small enough and large enough to satisfy both conditions.*
> >
> > Thank you for pointing this out. In our practical algorithm, we follow the implementation of CQL to introduce a ''budget'' parameter $\tau$ to automatically control $\varepsilon$ by dual gradient descent as discussed in Appendix B. It is mainly optimized for practical performance. Indeed, we cannot ensure that the resulting $\varepsilon$ satisfies both conditions. We will investigate this issue in our future study. However, we added further details in Appendix A.4 to show that a feasible $\varepsilon$ exists by choosing an approperiate $\alpha$.
> >
> > > *Were these hyperparameters optimized for the same environments used in your evaluation? It doesn't seem likely. If so, this is a huge flaw in the evaluation. How are the hyperparameters of PORL chosen?*
> >
> > In Appendix B, we described how we selected the hyperparameters of our approach.
> > A ''budget'' parameter $\tau$ was introduced to control $\epsilon$ in Eqn. 22, and the procedure of choosing $\tau$ was explained in the following paragraph.
> > As for the baselines, we found most original code bases had a set of configurations for the same or very similar enironments, and thus, we simlpy chose one of their configurations that led to the best performance.
> >
> > **Mid:**
> >
> > > *The problems associated with distributional shift, OOD states and actions, could be described better, e.g., in lines 24-26, it is claimed that distributional shift might cause problems, but what kind of problems is not well described (yet).*
> >
> > Thank you very much for your suggestion. We further elaborated the problems caused by distributional shift in lines 25-27.
> >
> > > *Neither $\mu$ nor its role are properly described. It is particularly confusing that $\mu$ is described to match the state-marginal, while in Eq.4, it appears as the state-conditional "policy" $\mu(a | s)$.*
> >
> > Thank you very much for your suggestion. The notation $\mu$ comes from the original formulation of CQL. Due to the space limit, we could not introduce the detailed derivaiton of CQL. In the revised version, to avoid confusion, we changed the notation $\mu$ to $\hat{\pi}^k$, which is the choice of $\mu$ in the practical implementation of CQL as well as in our algorithms.
> >
> > > *According to Eq.4, both CQL and PORL assumes access to the behavior policy $\pi_\beta$, which is a significantly stronger requirement compared to simple access to the dataset $\mathcal{D}$. Or is it simply the case that the dataset actions (which are sampled according to the behavior policy) are used directly? The details of how objective $J$ of Eq.7 is estimated should be provided (which expectations are computed exactly? which are approximated using a Monte Carlo estimation).*
> >
> > During the training process of offline reinforcement learning algorithms such as CQL and PORL, we only have access to the dataset $\mathcal{D}$. Therefore, the actions in the dataset are used directly.
> > As for the evaluation of the objective $J$ in Eqn. 7, we followed the convention in reinforcement learning community and approximated all expectations by Monte Carlo estimation. In the revised version, we add these details in the end of Sec. 5.2.

---

> > > ### Author Response · Authors · 2021-08-25
> > > **Response to Reviewer Qikv (Part 3/4)**
> > >
> > > (Continued. Part 3/4)
> > >
> > > > *in lines 124 and 125, the statement "it is empirically impossible to derive a solution to maximize the KL divergence" needs a citation or further justification.*
> > >
> > > Thank you very much for your suggestion. Actually, the KL divergence was only an example that introduced the following paragraphs. We did not intend to emphasize it in our paper. Therefore, we deleted it to avoid any confusion caused by it.
> > >
> > > > *In Eq.5, why only penalize a single action-value associated with unseen states, as opposed to all action-values associated with unseen states? I.e., why only use the action $a \sim \hat{\pi}^k(a|s)$?*
> > >
> > > Actually, the expections of the Q-function over the distribution $d^\phi(s)$ (or $d^{\pi_\beta}(s)$) and $\hat{\pi}^k(a|s)$ are penalized according to Eqn. 5.
> > > In the implementation of PORL, we use Monte Carlo esimation and sample batches of states and actions from the aforementioned distributions to approximate these expectations.
> > >
> > > > *Theorem 4.1 contains confusing statements which switch between $Q$ and $V$. "For any action and any state ..., i.e. $\hat{V}^\pi(s) \leq V^\pi(s)$...".*
> > >
> > > Thanks for pointing out the typo. We have modified the description in Theorem 4.1.
> > >
> > > > *In Theorem 4.1, the definition of $\hat{V}^\pi$ was unclear. I assume the intention was to define $\hat{V}^\pi$ as the convergence value $\hat{V}^\pi(s) = \lim_{k\rightarrow \infty} \hat{V}^k(s)$, but the way the sentence is worded it seems like there is an assumption that $\hat{V}^k$ converges to a specific value.*
> > >
> > > Thank you very much for your suggestion. We have changed definition of $\hat{V}^\pi$ to be the convergence value $\hat{V}^\pi(s) = \lim_{k\rightarrow \infty} \hat{V}^k(s)$ in Theorem 4.1.
> > >
> > > > *In line 384 of the appendix, "Setting the derivative to zero". The derivative with respect to what? It is not clear. Also, how is the derivative of the two regularization terms computed?*
> > >
> > > Thanks for pointing out the description that might cause confusion.
> > > We have added further details in Appendix A.1 to explain the derivation.
> > > Actually, the derivative is with respect to $Q$.
> > > The derivative of the regularization term is computed in the same way as other terms in Eqn. 5.
> > >
> > > > *In line 386 of the appendix, "It is obvious that the last two terms of Eqn. 8, ..., is indefinite". It is not obvious at all, and it is also unclear what the authors mean by "indefinite". It cannot be a real number? Since this is the appendix, the authors should feel free to use as much space as possible to be as clear as possible.*
> > >
> > > Thanks for pointing out the description that might cause confusion.
> > > We deleted ''indefinite'' and switched to a counter example.
> > > Basically, we would like to express that the last two terms of Eqn. 8 can be either positive or negative.
> > >
> > > > *In line 400 of the appendix, "Hence, we have ... $\hat{V}^{k+1} \leq \mathcal{B}^\pi \hat{V}^k$, which implies that each value-function update is a contraction." I am not sure if this is a typo, but this does not seem to align with the formal definition of contraction, which is an operator which guarantees the outputs associated with any two inputs are closer to each other than the inputs to each other. What do you mean by "contraction" here?*
> > >
> > > Thanks for point this out. We have modified the derivation of the fixed point in Appendix A.1 to include more details.
> > >
> > > > *Eq 13 in the appendix suddenly mixes point-wise and vector notation.*
> > >
> > > Thanks for your advice. We have rearranged and further explained the notations in Eqn. 13 to be more accurate about the vectors.

---

> > > > ### Author Response · Authors · 2021-08-25
> > > > **Response to Reviewer Qikv (Part 4/4)**
> > > >
> > > > (Continued. Part 4/4)
> > > >
> > > > **Minor:**
> > > >
> > > > > *PORL is an unfortunate acronym, since it conflicts with the widely used Partially Observable Reinforcement Learning (PORL).*
> > > >
> > > > Thank you very much for pointing this out! We really appreciate your suggestion and we have changed the abbreviation of pessimistic offline reinforcement learning from PORL to PessORL. However, we still used the name PORL in our response to avoid confusion.
> > > >
> > > > > *Calling Q-learning "dynamic-programming" (happens a few times) is inaccurate.*
> > > >
> > > > Thanks for your comment. Actually, we referred to [2] for catogarizing Q-learning into ''dynamic programming based'' methods.
> > > >
> > > > > *Notation of policy in line 84 does not work for discrete actions; should be $\delta[a = \arg \max_a ...]$.*
> > > >
> > > > Thanks for pointing this out. We have modified the notation as requested.
> > > >
> > > > > *$\mathcal{E}$ and $\mathcal{C}$ should be defined closer to their respective equations, i.e. $\mathcal{E}$ should be defined near Eq.1, and $\mathcal{C}$ should be defined near Eq.4.*
> > > >
> > > > Thanks for your advice. We have moved the definition of $\mathcal{E}$ below Eqn. 1, and the definition of $\mathcal{C}$ below Eqn. 4.
> > > >
> > > > > *The notation is at times a little bit redundant, e.g., $s \sim \mathcal{D}$ represents a state sampled from the dataset, and $s \sim d^{\pi_\beta}$ represents the same exact thing.*
> > > >
> > > > Thanks for pointing this out. We have modified the notation and have made sure that we stick to $s \sim d^{\pi_\beta}$ through out the paper after $d^{\pi_\beta}$ is defined in Sec. 4.1.
> > > >
> > > > > *Not accurate to use the same operator $\mathcal{B}^\pi$ for both state-values $V$ and action-values $Q$.*
> > > >
> > > > Thanks for your comment. Actually, the value function $V$ is the expectation of the state-action value function $Q$ over actions.
> > > > The bellman update operator $\mathcal{B}^\pi$ is invariant before and after taking the expection.
> > > > Therefore, we think it is acceptable to use $\mathcal{B}^\pi$ for $V$ and $Q$.
> > > >
> > > > > *Pessimistic MDPs should be formally described in the Background section.*
> > > >
> > > > The Pessimistic MDP was mentioned to elaborate the intuition of Theorem 4.2. We did not intend to emphasize the Pessimistic MDP in the paper, as it served only as an example. Therefore, we deleted the example about the pessimistic MDP.
> > > >
> > > >
> > > > [1] R. Kidambi, A. Rajeswaran, P. Netrapalli, and T. Joachims. ''Morel: Model-based offline323reinforcement learning.'' 2020.
> > > >
> > > > [2] S Levine, A Kumar, G Tucker, J Fu. ''Offline reinforcement learning: Tutorial, review, and perspectives on open problems.'' 2020.
> > > >
> > > > [3] A. Rajeswaran, V. Kumar, A. Gupta, G. Vezzani, J. Schulman, E. Todorov, and S. Levine. ''Learning complex dexterous manipulation with deep reinforcement learning and demonstrations,'' 2017.
> > > >
> > > > [4] A. Singh, A.Yu, J. Yang, J. Zhang, A. Kumar, and S. Levine. ''Cog: Connecting new skills to past experience with offline reinforcement learning,'' 2020.

---

### Meta-Review · Area_Chair_7BrL · 2021-08-06

**Recommendation:** Accept (Poster)
**Confidence:** 4

**Metareview:**

The reviewers generally found the paper interesting, well-motivated, and significant. I am happy that the authors used the rebuttal period efficiently, and have improved the submission in a number of ways based on reviewers' initial concerns.

---

> ### Author Response · Authors · 2021-08-25
> **Response to Area Chair 7BrL**
>
> Thank you very much for your time to read our paper and your valuable advice!
> We revised our paper according to the instructions and suggestions from the reviewers. Please find below the detailed responses to each reviewer.

---

### Decision · Program_Chairs · 2021-09-13

**Decision:**

Accept (Poster)

**Comment:**

The reviewers generally found the paper interesting, well-motivated, and significant. I am happy that the authors used the rebuttal period efficiently, and have improved the submission in a number of ways based on reviewers' initial concerns.